# Splat-MOVER: Multi-Stage, Open-Vocabulary Robotic Manipulation via Editable Gaussian Splatting

**Ola Shorinwa**[*1], **Johnathan Tucker**[*1], **Aliyah Smith**[2], **Aiden Swann**[2],
**Timothy Chen**[1], **Roya Firoozi**[1], **Monroe Kennedy III**[2], **Mac Schwager**[1]

[1]Dept. of Aeronautics and Astronautics, [2]Dept. of Mechanical Engineering, Stanford University

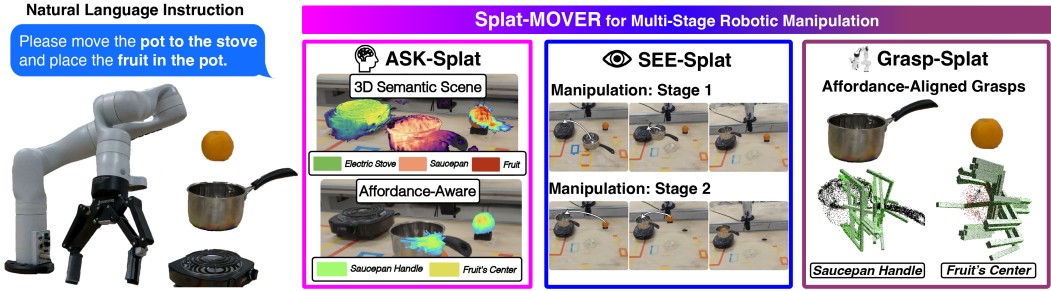

Figure 1: Splat-MOVER enables language-guided, multi-stage robotic manipulation, through an affordance-and-semantic-aware scene representation (ASK-Splat), a real-time scene-editing module (SEE-Splat), and a grasp-generation module (Grasp-Splat).

**Abstract:** We present Splat-MOVER, a modular robotics stack for open-vocabulary robotic manipulation, which leverages the editability of Gaussian Splatting (GSplat) scene representations to enable multi-stage manipulation tasks. Splat-MOVER consists of: (i) *ASK-Splat*, a GSplat representation that distills semantic and grasp affordance features into the 3D scene. ASK-Splat enables geometric, semantic, and affordance understanding of 3D scenes, which is critical for many robotics tasks; (ii) *SEE-Splat*, a real-time scene-editing module using 3D semantic masking and infilling to visualize the motions of objects that result from robot interactions in the real-world. SEE-Splat creates a "digital twin" of the evolving environment throughout the manipulation task; and (iii) *Grasp-Splat*, a grasp generation module that uses ASK-Splat and SEE-Splat to propose affordance-aligned candidate grasps for open-world objects. ASK-Splat is trained in real-time from RGB images in a brief scanning phase prior to operation, while SEE-Splat and Grasp-Splat run in real-time during operation. We demonstrate the superior performance of Splat-MOVER in hardware experiments on a Kinova robot compared to two recent baselines in four single-stage, open-vocabulary manipulation tasks. In addition, we demonstrate Splat-MOVER in four multi-stage manipulation tasks, using the edited scene to reflect changes due to prior manipulation stages, which is not possible with existing baselines. Video demonstrations and the code for the project are available at https://splatmover.github.io.

**Keywords:** Gaussian Splatting, Robotic Manipulation, Scene Editing

## 1 Introduction

Open-world robotic manipulation requires spatial and semantic understanding of the scene in which a robot is operating. In particular, a robot must be able to identify the location and geometry of objects in its environment based on their semantic attributes. However, spatial and semantic scene

---

[*]Equal contribution. Correspondence to {shorinwa, jatucker}@stanford.edu.

8th Conference on Robot Learning (CoRL 2024), Munich, Germany.

understanding alone is often inadequate in robotic manipulation. In many cases, it is critical for the robot to know the best place to grasp the object to perform the required task, that is, to be able to detect and localize grasp affordances on the object [1, 2]. Furthermore, for multi-stage manipulation tasks, where the pre-conditions for the next stage are met through the success of a previous stage, it is essential to continually update the 3D scene model to reflect the objects' motions due to the robot's previous actions, to have an accurate scene representation that is useful for subsequent stages.

Consequently, in this work, we introduce Splat-MOVER, a modular robotics stack for **M**ulti-Stage, **O**pen-**V**ocabulary **R**obotic Manipulation via **E**ditable Gaussian Splatting. Splat-MOVER consists of three modules: (i) a 3D **A**ffordance-and-**S**emantic **K**nowledge Gaussian Splatting scene representation (ASK-Splat), (ii) a **S**cene-**E**diting-**E**nabled module for Gaussian Splatting scenes (SEE-Splat), and (iii) a grasp generation module (Grasp-Splat) that uses ASK-Splat and SEE-Splat to plan grasps in multi-stage manipulation tasks. These three modules together make up Splat-MOVER, illustrated in Fig. 1. In a brief pre-scanning phase, we train the ASK-Splat model from posed RGB images of the workspace, simultaneously embedding CLIP and grasp affordance features from the images into the 3D scene model. Then, given a natural language prompt of a multi-stage manipulation task, Splat-MOVER uses ASK-Splat to localize and mask the objects from the prompt in the 3D scene, providing the object's 3D initial and target configurations at each stage of the task. Subsequently, SEE-Splat edits the ASK-Splat scene, reflecting the dynamic poses of the objects throughout the manipulation task. Finally, Grasp-Splat uses the pre-trained GraspNet model [3] to generate candidate grasps, which are then re-ranked based on their affordance scores from ASK-Splat, to obtain grasp poses for each object, at each stage of the manipulation task. Point-to-point motion planning for the robot arm is accomplished with standard off-the-shelf planning tools, e.g., MoveIt [4].

We showcase Splat-MOVER's effectiveness through hardware experiments on a Kinova robot, where Splat-MOVER achieves significantly improved success rates across four single-stage open-vocabulary manipulation tasks compared to two recent baseline methods: LERF-TOGO [5] and F3RM* [6]. In three single-stage manipulation tasks, Splat-MOVER improves the success rate of LERF-TOGO by a factor of at least $2.4$, while achieving similar success rates in a fourth task ($95\%$ compared to LERF-TOGO's $100\%$). Likewise, Splat-MOVER outperforms F3RM* by a factor ranging from $1.2$ to $3.3$, across the four tasks. In addition, we demonstrate the performance of Splat-MOVER in four multi-stage manipulation tasks, where we leverage SEE-Splat to reflect the updates in the scene resulting from prior manipulation stages, a capability absent in existing baseline approaches.

## 2 Preliminaries

We introduce notation relevant to this paper, before presenting Gaussian Splatting. We denote a 2D feature embedding as $f \in \mathbb{R}^{a \times b \times c}$, where the first-two dimensions $(a, b)$ represent the spatial dimensions, while the last dimension $c$ represents the dimension of the embedding. Similarly, we denote 1D feature embeddings as $f \in \mathbb{R}^c$. We provide a brief review of Gaussian Splatting, which we build upon to obtain a affordance-and-semantic-aware scene representation. We direct interested readers to [7] for a more in-depth discussion of Gaussian Splatting. In Gaussian Splatting, the scene is represented using 3D Gaussians, each parameterized by a mean $\mu \in \mathbb{R}^3$ and covariance $\Sigma \in \mathbb{R}^{3 \times 3}$, which is represented as the product of a scaling (diagonal) matrix $S \in \mathbb{R}^{3 \times 3}$ and a rotation matrix $R \in \mathbb{R}^{3 \times 3}$. Each Guassian has an opacity parameter $\alpha \in \mathbb{R}$ and spherical harmonics parameters for view-dependent visual properties. Tile-based rasterization with $\alpha$-blending is utilized for rendering the Gaussians. The parameters of each Gaussian are optimized through stochastic gradient descent, leveraging the differentiability of the tile-based rasterization procedure to compute the gradients of the loss function $\mathcal{L}$, which consists of an $\ell_1$-photometric loss and a similarity (SSIM) loss [8].

## 3 Affordance-and-Semantic-Knowledge Gaussian Splatting

Given a dataset $\mathcal{D}$ of RGB images, given by $\mathcal{D} = \{\mathcal{I}_1, \ldots, \mathcal{I}_N\}$, ASK-Splat embeds: (1) CLIP features, and (2) grasp affordance features in the GSplat scene. Using the CLIP features, ASK-Splat

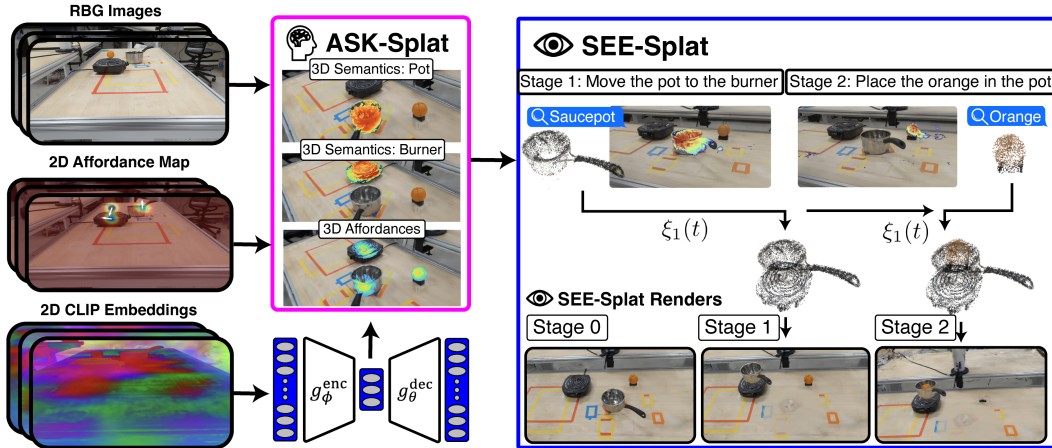

Figure 2: ASK-Splat grounds 2D visual attributes (e.g, color and lighting effects), grasp affordance, and semantic embeddings within a 3D GSplat representation and is trained entirely from RGB images. Using 3D ASK-Splat, SEE-Splat enables open-vocabulary scene-editing via semantic localization of Gaussian primitives in the scene, followed by 3D masking and transformation $\xi(t)$ of these Gaussians.

generates a 3D heatmap given an open-vocabulary text prompt, which denotes the semantic relevance to the prompt. E.g., the prompt "apple" will give a 3D heat map that is hottest over the apple, but also somewhat hot over the pear and orange (which are semantically "close" to apple). Although the affordance feature field cannot be queried directly in natural language, it provides information on the graspability of all points in the scene. We use the CLIP features to mask out relevant objects and the affordance features to highlight graspable locations, as illustrated in Fig. 2.

**Grounding Language Semantics in 3D Gaussian Splatting.** We distill *ground-truth* image embeddings from the vision-language foundation model CLIP, denoted by $f_{\text{gt}} \in \mathbb{R}^{H_f \times W_f \times C}$, extracted from $\mathcal{D}$ via MaskCLIP [9], into the 3D scene. To preserve the real-time rendering speed of Gaussian Splatting, we learn a latent space for the CLIP embeddings with an autoencoder, consisting of an encoder $g_\phi^{\text{enc}} : \mathbb{R}^{H_f \times W_f \times C} \mapsto \mathbb{R}^{H_f \times W_f \times l}$ and a decoder $g_\theta^{\text{dec}} : \mathbb{R}^{H_f \times W_f \times l} \mapsto \mathbb{R}^{H_f \times W_f \times C}$, compressing the semantic feature space of dimension $C$ to the semantic latent space of dimension $l$ and vice-versa, with parameters $\phi$ and $\theta$, respectively, depicted in Fig. 2. We set $l = 3 \ll C$. The autoencoder is trained to minimize the reconstruction loss.

We distill the latent CLIP features into the GSplat scene, by augmenting the attributes of each Gaussian with an additional parameter $f_s \in \mathbb{R}^l$, denoting the semantic feature of the Gaussian, which are optimized via gradient descent on the loss function:

$$\mathcal{L}_s = \kappa_s \sum_{i=1}^{|\mathcal{D}|} \left\| \hat{\mathcal{I}}_i^f - g_\phi^{\text{enc}}(f_{\text{gt},i}) \right\|_2^2 + \frac{1}{|\mathcal{D}|} \sum_{i=1}^{|\mathcal{D}|} \left( 1 - \psi(\hat{\mathcal{I}}_i^s - g_\phi^{\text{enc}}(f_{\text{gt},i})) \right), \tag{1}$$

where $\hat{\mathcal{I}}_i^s \in \mathbb{R}^{H \times W \times l}$ denotes the 2D semantic feature map rendered from the Gaussian Splats, $\psi : \mathbb{R}^{U \times V \times C} \times \mathbb{R}^{U \times V \times C} \mapsto \mathbb{R}$ denotes the cosine-similarity function (where we note that $\psi$ applies to inputs of arbitrary height and width), and $\kappa_s \in \mathbb{R}_{++}$ denotes the constant term in the mean-squared-error (MSE) loss. To generate a semantic similarity map given a natural-language query (*positive* query), we compute the cosine-similarity between the CLIP embedding associated with the query and the distilled semantic embeddings associated with each Gaussian in the scene. We transform the computed cosine-similarity to a probability using the pairwise softmax function (given a *negative* query), which is thresholded to generate the semantic similarity map. We describe the distillation and optimization procedures in greater detail in Appendix A.

**Grounding Affordance in 3D Gaussian Splatting.** ASK-Splat embeds object-specific grasp affordances directly within the 3D Gaussian Splatting environment, yielding a heat map over objects in the scene relating to the graspability. We distill visual grasp affordances from a vision-affordance

foundation model trained on large human-object interaction datasets, VRB [10], into the scene representation, although we note that other vision-affordance foundation models can also be used. The distilled affordance endows robots with the ability to identify the parts of an object that a human is more likely to grasp, capturing common-sense human knowledge and experience in interacting with objects. We obtain the *ground-truth* affordance score by evaluating the training dataset $\mathcal{D}$ using VRB and leverage the 3D Gaussian primitives in embedding the 2D visual grasp affordance scores in 3D. By augmenting the attributes of each Gaussian in the scene with an additional parameter $\beta \in [0, 1]$, representing the *score* of the Gaussian affording the task of grasping, we ground the grasp affordances within the geometric primitives representing the occupied regions of the scene. We enforce the box constraints on $\beta$ using the sigmoid activation function to provide smooth gradients during the training procedure. We provide additional details in Appendix A.

## 4   Scene-Editing-Enabled Gaussian Splatting

SEE-Splat consists of three components: (i) a semantic component utilizing natural-language queries to generate a relevancy map based on semantic similarity; (ii) a Gaussian masking module that generates a sparse point cloud of the relevant objects, comprising of all semantically-relevant Gaussians; and (iii) a scene transformation component that edits the 3D scene by inserting, removing, or modifying the geometric, spatial, or visual properties of the Gaussians. Together, these components enable robots to visualize the effects of their interactions with other objects in a virtual 3D scene, prior to executing these actions in the real-world. In Fig. 2, we illustrate each component of SEE-Splat by editing a real-world cooking scene to visualize the action of moving the saucepot from the table to the electric stove, showing localization of the saucepot, extraction of the semantically-relevant Gaussians, and transformation of the relevant Gaussians.

**Semantic Localization via ASK-Splat.** Given a natural-language query specifying an object of interest, SEE-Splat leverages ASK-Splat to identify semantically relevant Gaussians, generating a semantic similarity map. To improve the localization accuracy, the text prompt can include the geometric and visual properties of the object, such as its color, in addition to its semantic class.

**Masking the Gaussians in SEE-Splat.** Given a semantic similarity map of the scene, SEE-Splat generates a mask identifying the Gaussians relevant to the specified object. Given a threshold for semantic relevancy, SEE-Splat removes dissimilar Gaussians from the scene, creating a sparse point cloud of the relevant Gaussians, constructed from the means of these Gaussians. However, photo-realistic rendering of Gaussian environments require denser point clouds. To densify the point cloud, SEE-Splat lifts the features of the point cloud from the 3D Euclidean space to a 7D feature space, by augmenting each point in the point cloud with its RGB color and semantic score, and subsequently identifies all neighboring Gaussians in the scene within a specified distance of the point cloud in the 7D feature space using an efficient KD-tree query. SEE-Splat incorporates these points into the point cloud to create a denser point cloud, comprising of all semantically-relevant Gaussians, while removing outliers from the set of points.

**Editing the Gaussians in SEE-Splat.** To edit ASK-Splat, SEE-Splat applies a transform $\xi : \mathcal{G}_s \mapsto \mathcal{G}_s$ to the relevant Gaussians to update their spatial and geometric attributes, where $\mathcal{G}_s$ represents the space of the Gaussian primitives. The transform $\xi$, which is not limited to rigid transformations, can be generated via heuristics, a large language model (LLM), a human, or computed from the observations collected by onboard sensors. With these capabilities, SEE-Splat provides a digital twin of the scene. Moreover, provided sensor feedback is available, SEE-Splat enables real-time visualization of the real-world in a virtual environment, enabling the Gaussian primitives to accurately reflect the real-time geometry and visual properties of objects in the real-world. Although, we do not consider physics-based simulations in this work, we note that physics can be incorporated into SEE-Splat to achieve greater realism. We expound on this point in our discussion on the limitations of SEE-Splat. In addition, while editing the scene, SEE-Splat removes visual and geometric artifacts through 3D Gaussian infilling, shown in Fig. 8, which we discuss in detail in Appendix B.

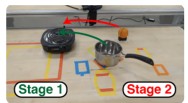 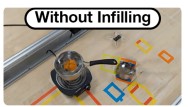 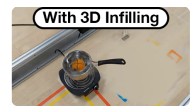  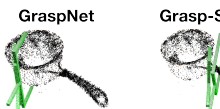

Figure 3: Scene-editing introduces artifacts, e.g., holes in the table (center) after moving the saucepan, which are removed via 3D Gaussian infilling (right).

Figure 4: The top-two grasps proposed by (left) GraspNet and (right) Grasp-Splat for a saucepan.

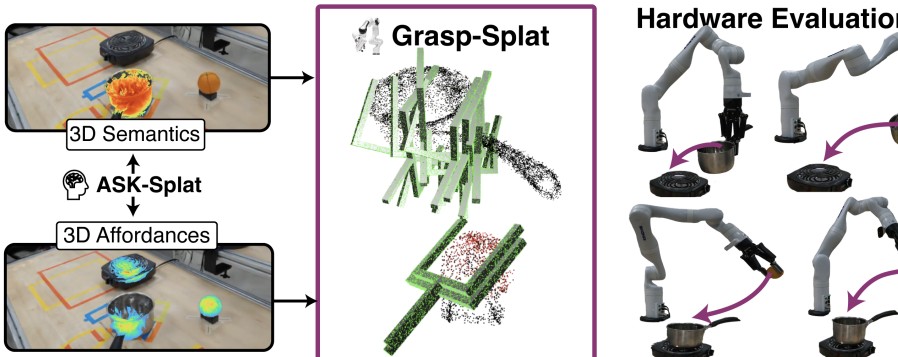

Figure 5: Grasp-Splat generates affordance-aligned grasps using the semantic and grasp-affordance knowledge in ASK-Splat. Qualitatively, the proposed grasps lie in regions where a human is more likely to grasp (e.g., the handle of the saucepot and the center of the fruit) and are more likely to result in stable grasps, when executed by a robot.

## 5 Affordance-Aligned Grasp Generation

Grasp-Splat utilizes the dense point cloud of the object generated by SEE-Splat to propose grasp configurations for grasping the object, while harnessing the grasp affordance knowledge in ASK-Splat to identify generally more stable grasp configurations associated with the specified object. The proposed grasps are generated from the point cloud using a deep-learned model GraspNet [3], that estimates 6D grasp poses for a parallel-jaw gripper from a point cloud of the object, along with estimated grasp-quality scores associated with these grasp poses. We note that the grasps generated by GraspNet are not always ideal (see Fig. 4). Consequently, Grasp-Splat executes a grasp selection procedure to identify more-promising candidate grasps. We hypothesize that leveraging grasp affordance of each part of the object when generating candidate grasps could be essential to identifying better candidate grasps. As a result, we introduce the grasp metric $\nu : \mathrm{SE}(3) \mapsto \mathbb{R}$, which computes the grasp affordance at a specified grasp pose $X \in \mathrm{SE}(3)$. Grasp-Splat ranks the candidate grasps generated by GraspNet based on the grasp scores given by $\nu(X)$, leveraging the affordance associated with each grasp pose to identify grasp configurations that are more likely to succeed, depicted in Fig. 9. Moreover, since GraspNet does not consider the position of the robot relative to the object, the proposed grasps might require post-processing to account for the relative position of the robot. We provide additional discussion in Appendix C.

## 6 Experiments

We compare Splat-MOVER to the existing open-vocabulary robotic manipulation methods LERF-TOGO [5] and F3RM [6] in four tasks: the *Cooking* task, *Chopping* task, *Cleaning* task, and *Workshop* task, illustrated in Table 2, executed on a Kinova robot. We utilize the implementation of LERF-TOGO provided by the authors. We refer to our implementation of F3RM [6] using GraspNet for grasp generation in place of the virtual-reality-collected task demonstrations as F3RM*. We summarize the capabilities of each method in Table 1 and provide additional results in Appendix D.

| Capabilities/Methods | Semantics | Affordance | Scene-Editing |
|---|---|---|---|
| LERF-TOGO [5] | ✓ | ✗ | ✗ |
| F3RM* [6] | ✓ | ✗ | ✗ |
| Splat-MOVER (**ours**) | ✓ | ✓ | ✓ |

Table 1: Representation Capabilities of LERF-TOGO [5], F3RM* [6], and Splat-MOVER

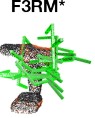 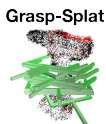

Figure 6: Grasps proposed by GraspNet, LERF-TOGO, F3RM*, and Grasp-Splat for a *powerdrill*.

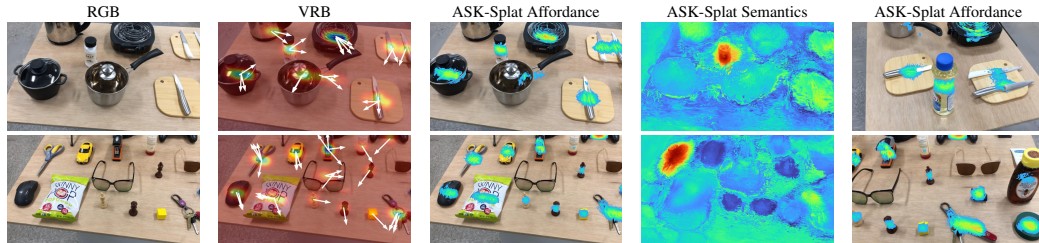

Figure 7: Semantic and Affordance Knowledge encoded by ASK-Splat. We show the RGB image, grasp affordance as predicted by the vision-affordance foundation model (VRB), the grasp affordance from ASK-Splat from rendered RGB images at novel views and the semantic similarity map for a *salt shaker* and a pair of *scissors*.

**ASK-Splat Representation.** We evaluate the grasp affordance and semantic knowledge embedded in ASK-Splat. In Figure 10, we show the RGB image, grasp affordance heatmap computed by VRB, and the grasp affordance heatmap rendered from ASK-Splat composited with the rendered RGB image, in addition to the semantic similarity map for a *salt shaker* and a pair of *scissors*. The affordance heatmap shows the regions in each object amenable to grasping. Qualitatively, from Figure 10, ASK-Splat encodes the grasp affordance given by VRB, identifying reasonable regions on each object for grasping. Although VRB provides the 2D motion direction associated with each grasp affordance region, we do not distill this knowledge into ASK-Splat, as we found the 2D motion directions to be quite noisy and relatively uninformative.

**Single-Stage Manipulation.** We consider grasping a saucepan, knife in a knife guard, cleaning spray, and power drill. We show the grasps proposed by each method for a power drill in Fig. 12. In the hardware experiments on the robot, we describe a candidate grasp as being feasible if the MoveIt planner successfully finds a plan to execute the grasp. We execute 20 feasible trials for each object, and in Table 2, provide the grasping success rate of each method, as well as the percentage of grasps that lie within the affordance region of the object (AGSR), where the affordance region of each object is defined as follows: the entire handle of the saucepan, the entire knife with the guard on, the region of the cleaning spray excluding its cap, and the middle region of the power drill below its top compartment and above its battery compartment.

From Table 2, Splat-MOVER achieves the best grasping success rates and the highest percentage of grasps in the affordance region in grasping the saucepan, knife, and cleaning spray. LERF-TOGO attains a perfect success rate in grasping the power drill; however, LERF-TOGO always grasped the top of the drill, outside its affordance region, shown in Appendix D. In contrast, Splat-MOVER achieves a 95% success rate, grasping the object within its affordance region. Splat-MOVER outperforms the other methods in the first stage of the task likely due to the embedded grasp affordance in ASK-Splat. To generate grasps, Splat-MOVER exploits the distilled grasp affordance in ASK-Splat, which encodes common-sense knowledge of more suitable grasp locations on an object, drawn from human experience. In contrast, LERF-TOGO generates candidate grasp conditioned on a text prompt identifying the region to grasp the object (such as its handle) provided by a human operator or an LLM, while and F3RM* does so from a dataset of human demonstrations. We note that the semantic part description can introduce ambiguity in tasks where the "handle" of the object is not well-defined, such as the cleaning spray.

In addition, we evaluate the pick-and-place success rate of all the methods in each manipulation task, where the place success rate is conditioned on the number of successful trials in picking the object. The publicly-available implementation of LERF-TOGO does not support the specification of a place location. Hence, we do not evaluate its place success rate. In Table 2, we present the pick-and-place success rates in the four tasks. In the Cooking task, Splat-MOVER achieves a perfect success rate in picking up the saucepan, with a place success rate of 60%. We note that the place step involves placing the saucepan on an electric burner, which is quite challenging, potentially explaining the lower place success rate. Nonetheless, Splat-MOVER outperforms LERF-TOGO and F3RM* in the first stage of the task, the only stage of the task that both methods can perform.

**Multi-Stage Manipulation.** Since the NeRF-based methods LERF-TOGO and F3RM* are not amenable to multi-stage manipulation, we cannot evaluate the success rate of these methods for the entire manipulation task. In contrast, Splat-MOVER enables multi-stage robotic manipulation, achieving minimum pick-and-place success rates of 50% and 80%, respectively, across the Cooking, Chopping, Cleaning, and Workshop tasks, as shown in Table 2.

Table 2: Grasping success rate, the percentage of grasps in the affordance region (AGSR), and the pick and place success rates for the two-stage manipulation tasks across 20 feasible trials for a *Cooking* task; *Chopping* task; *Cleaning* task; and *Workshop* task. (The best-performing stat is in bold.)

| | Methods | Saucepot | | Stage 1 | | Stage 2 | |
| | | Grasp (%) | AGSR (%) | Pick (%) | Place (%) | Pick (%) | Place (%) |
|---|---|---|---|---|---|---|---|
|  | LERF-TOGO [5] | 40 | 5 | 40 | N/A | N/A | N/A |
| | F3RM* [6] | 30 | 30 | 30 | 0 | N/A | N/A |
| | Splat-MOVER (**ours**) | **100** | **60** | **100** | **60** | **50** | **80** |
| | Methods | Knife | | Stage 1 | | Stage 2 | |
| | | Grasp (%) | AGSR (%) | Pick (%) | Place (%) | Pick (%) | Place (%) |
|  | LERF-TOGO [5] | 35 | 35 | 35 | N/A | N/A | N/A |
| | F3RM* [6] | 60 | 60 | 60 | **91.67** | N/A | N/A |
| | Splat-MOVER (**ours**) | **85** | **85** | **85** | 82.35 | **65** | **100** |
| | Methods | Cleaning Spray | | Stage 1 | | Stage 2 | |
| | | Grasp (%) | AGSR (%) | Pick (%) | Place (%) | Pick (%) | Place (%) |
|  | LERF-TOGO [5] | 25 | 15 | 25 | N/A | N/A | N/A |
| | F3RM* [6] | 75 | 40 | 75 | 6.67 | N/A | N/A |
| | Splat-MOVER (**ours**) | **90** | **90** | **90** | **83.33** | **70** | **100** |
| | Methods | Powerdrill | | Stage 1 | | Stage 2 | |
| | | Grasp (%) | AGSR (%) | Pick (%) | Place (%) | Pick (%) | Place (%) |
|  | LERF-TOGO [5] | **100** | 0 | **100** | N/A | N/A | N/A |
| | F3RM* [6] | 70 | 70 | 70 | 7.14 | N/A | N/A |
| | Splat-MOVER (**ours**) | 95 | **95** | 95 | **94.74** | **85** | **88.24** |

# 7   Related Work

**Open-World Robotic Manipulation.** Recent research has trained foundation models for end-to-end visuomotor control, enabling open-world robotic manipulation built off the vast knowledge encoded within the foundation models [11, 12, 13, 14, 15]. Rather than directly training a foundation model, other works have distilled the knowledge from these foundation models into a map representation, constructing 3D feature fields, with semantic embeddings enabling open-vocabulary language-guided robotic manipulation [5] [6]. Despite these advances, open-world, end-to-end visuomotor policies generally require expensive fine-tuning procedures, while existing 3D feature fields are limited to single-stage manipulation tasks and require human demonstrations or external guidance from an LLM to generate feasible grasps for the manipulation task. Addressing some of these limitations, Splat-MOVER enables multi-stage manipulation tasks via real-time scene-editing and embeds grasp-affordance and semantic features within the learned scene, rendering external guidance unnecessary.

**Language-Embedded NeRFs and Gaussian Splats.** Neural radiance fields (NeRFs) [16] have enabled high-fidelity scene representations, with view-consistent 3D geometry, yielding photorealistic renderings, with many robotic applications [17, 18, 19, 20, 21, 22, 23]. Many works [24, 25, 26, 9, 25] have leveraged NeRFs to ground semantic knowledge from 2D vision-language foundation models [27, 28, 29] in 3D to enable consistent object masking via natural-language queries. More recently, Gaussian Splatting [7] has emerged as an effective 3D scene representation, providing notable competitive advantages compared to NeRFs, including real-time rendering speeds and greater flexibility for dynamic scene-editing. Splat-MOVER leverages these competitive features to provide a digital twin of a scene, enabling multi-stage manipulation. As in the case with NeRFs, recent works [30, 31, 32, 33, 34] have embedded semantic features into 3D Gaussian Splatting, often requiring comparatively intensive training procedures. In comparison, Splat-MOVER introduces a lightweight technique for semantic distillation and applies the resulting model specifically to robotic manipulation. We provide a more extensive discussion of the related work in Appendix E.

**Grasp Generation in Gaussian Splatting Scenes.** In concurrent work, GaussianGrasper [35] leverages the deep-learned model AnyGrasp [36] to generate robotic grasps from GSplat environments. However, GaussianGrasper does not embed grasp affordances or consider scene editing for multi-stage tasks, as we do. ManiGaussian [37] uses a GSplat scene to represent the robot arm itself, and predicts optimal arm actions for grasping, given this representation. The method is shown in simulation only. In addition, the workshop paper [38] considers a GSplat method for tracking objects during robot manipulation motions, but does not consider robotic planning or distilled affordances, as is our focus here. Finally, GraspSplats [39] utilizes the sampling-based grasp-generation method GPG [40] to propose grasp configurations and utilizes RGB-D inputs for depth supervision and real-time object tracking, whereas our work focuses on multi-stage robotic manipulation.

## 8 Conclusion

We present Splat-MOVER, a robotics stack for multi-stage open-vocabulary robotic manipulation, consisting of three modules: ASK-Splat, SEE-Splat, and Grasp-Splat. ASK-Splat enables semantic and affordance queries via natural-language interaction to identify relevant objects within 3D scenes, while SEE-Splat provides real-time, dynamic scene editing, enabling visualization of the evolution of the real-world scene due to the robot's interaction with objects within the scene. Grasp-Splat builds upon these two modules for affordance-aware grasp generation, necessary for effective multi-stage robotic manipulation. We demonstrate the effectiveness of Splat-MOVER in real-world experiments in comparison to two recent baseline methods. Compared to the existing works, Splat-MOVER endows robots with the unique capability for multi-stage, open-vocabulary manipulation with minimal human inputs, by leveraging distilled grasp affordance knowledge and real-time dynamic scene editing, which are essential to multi-stage manipulation.

## 9 Limitations and Future Work

We present a few limitations of Splat-MOVER and provide directions for future work. ASK-Splat distills grasp affordance knowledge from foundation models into 3D Gaussian Splatting scenes. We note that the effectiveness of the distilled grasp affordances and the generalization capability of our model is highly dependent on that of the affordance foundation model. For example, when trained using the VRB foundation model, an ASK-Splat scene may not generalize notably well to markedly-different objects outside of those seen in the EPIC-KITCHENS dataset. Future work will examine enhancing the generalization capability of the proposed ASK-Splat module by training and employing diverse cross-environment, cross-task, vision-affordance foundation models, rather than relying on VRB, which is specifically trained for kitchen tasks. Further, in its current form, the distilled visual grasp affordance does not depend on the orientation of the candidate grasp, limiting Grasp-Splat from fully harnessing grasp affordances in identifying better grasp configurations in $SE(3)$. Future work will consider extending the computation of grasp affordance to $SE(3)$. We present additional directions for future work in Appendix F.

**Acknowledgments**

This work was supported in part by DARPA grant HR001120C0107, NSF Graduate Research Fellowship DGE-1656518 and DGE-2146755, NSF Grants 2220867 and 2342246, ONR grant N00014-23-1-2354, and by a gift from Meta. We are grateful for this support. Toyota Research Institute provided funds to support this work.

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

# Appendix A    Affordance-and-Semantic-Knowledge Gaussian Splatting

Here, we provide additional details of the distilled knowledge in ASK-Splat.

## A.1    Grounding Language Semantics in $3$D Gaussian Splatting

A naive approach to encode semantic information within a Gaussian Splatting representation would be to assign an additional parameter $f \in \mathbb{R}^C$ assigned to each Gaussian, representing the semantic feature embedding associated with the Gaussian. However, such an approach introduces significant memory and computation challenges, effectively eliminating the real-time rendering rates of Gaussian Splatting. To preserve the real-time rendering speed of Gaussian Splatting, we learn a lower-dimensional latent space associated with the high-dimensional semantic feature embeddings, and subsequently, leverage the lower-dimensional latent features for semantic knowledge distillation. We note that although the image embeddings from [9] are sometimes described to be dense, the resulting image embeddings for the original input image are essentially patch-based, with $H_f < H$ and $W_f < W$. Further, we note that the dimension of $f_{\text{gt}}$ (denoted by $C$) depends on the CLIP model used, with common values ranging between $512$ (for the CLIP-ViT models) and $1024$ (for the CLIP-ResNet models).

In this work, we utilize feedforward MLPs in defining the encoder and decoder with $l = 3 \ll C$. However, we note that larger values of $l$ generally result in more expressive semantic scene representations, at the expense of increased memory and rendering costs. We train the autoencoder with the loss function $\mathcal{L}_g$, given by:

$$\mathcal{L}_g = \kappa_g \sum_{i=1}^{|\mathcal{D}|} \left\| g_\theta^{\text{dec}}(g_\phi^{\text{enc}}(f_{\text{gt},i})) - f_{\text{gt},i} \right\|_2^2 + \frac{1}{|\mathcal{D}|} \sum_{i=1}^{|\mathcal{D}|} \left( 1 - \psi(g_\theta^{\text{dec}}(g_\phi^{\text{enc}}(f_{\text{gt},i})), f_{\text{gt},i}) \right), \qquad (2)$$

where $g_\theta^{\text{dec}}(g_\phi^{\text{enc}}(\cdot))$ represents the composition of the encoder and decoder that outputs the reconstruction of its inputs, $\psi : \mathbb{R}^{U \times V \times C} \times \mathbb{R}^{U \times V \times C} \mapsto \mathbb{R}$ denotes the cosine-similarity function (where we note that $\psi$ applies to inputs of arbitrary height and width), and $\mathcal{D}$ denotes the dataset of images used in training the Gaussian Splatting scene, with $f_{\text{gt},i}$ denoting the ground-truth semantic features of image $i$. The first term in (2) represents the mean-squared-error (MSE) reconstruction loss with $\kappa_g \in \mathbb{R}_{++}$ denoting the constant associated with this term, while the second term represents the cosine-similarity loss between the ground-truth embeddings and the reconstruction. Subsequently, we distill the lower-dimensional embeddings into the Gaussian Splatting representation, resizing the output of $g_\phi^{\text{enc}}$ using bilinear interpolation to obtain a ground-truth semantic map of compatible dimensions.

Given a good initialization of the Gaussians (e.g., when the sparse point cloud from structure-from-motion is utilized in initializing the Gaussians), the semantic feature parameters associated with each Gaussian can be trained simultaneously with other spatial and visual-related parameters associated with the Gaussians, along with the autoencoder's parameters. Nonetheless, empirical evaluations suggest that a sequential training procedure in which the semantic parameters are trained after the non-semantic parameters of the Gaussians have been trained yields better localized semantic feature maps. We hypothesize that this observation may result from more consistent grounding of the semantic features.

To evaluate the semantic similarity between a specified query and the objects in the scene, we compute the text embedding of the query using CLIP [27] and utilize the cosine-similarity metric, which is widely used in prior work [26]. Consistent with prior work, we allow for the specification of *negative* queries to help distinguish between dissimilar objects and the object of interest described by a *positive* query. However, we note that, in practice, a positive query suffices without negative queries. We compute the embeddings of each item in the set of negative queries and positive queries, and subsequently compute the cosine similarity between the predicted semantic feature given by the Gaussian Splats and each query in the set of negative and positive queries. Finally, the similarity

score between a feature point $p$ and the positive query is given by:

$$\text{sim}(\mathcal{Q}_s, \mathcal{Q}'_s) = \min_{i \in |\mathcal{Q}'_s|} \gamma(p, \nu_a(\mathcal{Q}_s), \nu_b(q'_{s,i})), \tag{3}$$

where $\mathcal{Q}$ denotes the set of positive queries, $\mathcal{Q}'_s = \{q'_{s,i}, \ \forall i \in [|\mathcal{Q}'_s|]\}$ denotes the set of negative queries, $\nu_a : \mathcal{S} \mapsto \mathbb{R}$ computes the average semantic embedding of a set of text prompts $\bar{s} \in \mathcal{S}$, $\nu_b(q'_{s,i})$ represents the semantic embedding of the negative query $q'_{s,i}$, and $\gamma : \mathbb{R}^3 \times \mathbb{R}^C \times \mathbb{R}^C \mapsto \mathbb{R}$ represents the pairwise softmax function over the positive query embedding and the $i$th negative query embedding at the 3D feature point, outputting the probability associated with the positive query embedding. In general, when rendering the semantic similarity maps, we apply a threshold of $0.5$ to the similarity score computed in (3) to distinguish feature points associated with the query from dissimilar ones.

### A.2 Grounding Affordance in $3$D Gaussian Splatting

We embed grasp affordances in ASK-Splat. During training, we utilize the same tile-based rasterization procedure discussed in Section A.1 in rendering the $2D$ visual grasp affordance of the scene and optimize the grasp affordance parameter $\beta$ using the loss function: $\mathcal{L}_\beta = \kappa_\beta \sum_{i=1}^{|\mathcal{D}|} \left\| \hat{\mathcal{I}}_i^\beta - \mathcal{I}_i^\beta \right\|_2^2$, which represents the MSE loss between the ground-truth $2D$ visual grasp affordance $\mathcal{I}_i^\beta \in \mathbb{R}^{H \times W \times 1}$ and the rendered visual grasp affordance $\hat{\mathcal{I}}_i^\beta \in \mathbb{R}^{H \times W \times 1}$. We optimize the affordance parameters concurrently with the non-semantic parameters of each Gaussian. From a trained ASK-Splat scene, we can generate dense 2D visual grasp affordance maps, as well as sparser 3D visual grasp affordance maps, by directly evaluating the affordance score associated with each Gaussian.

## Appendix B  Scene-Editing-Enabled Gaussian Splatting

We provide additional details on the Gaussian-editing component of SEE-Splat.

### B.1 Editing the Gaussians in SEE-Splat

Deletion and transformation of the Gaussians introduces artifacts into the scene representation, degrading its photo-realistic qualities. To address this challenge, SEE-Splat enables 3D Gaussian infilling by introducing new Gaussians with similar attributes in regions with missing geometry. Figure 8 provides an illustration of such artifacts (e.g., the hole in the table), when the scene is edited to visualize the effects of moving the saucepan to the electric stove. Through 3D Gaussian infilling, SEE-Splat generates a photorealistic rendering of the edited scene, eliminating these artifacts.

## Appendix C  Grasping and Manipulation with Splat-MOVER

We present Grasp-Splat and discuss its application to multi-stage robotic manipulation via Splat-MOVER.

### C.1 Grasp-Splat for Grasp Proposal

We note that the grasps generated by GraspNet are not always ideal. For example, the grasps generated by GraspNet in Figure 9 are either infeasible or challenging to execute. As a result, Grasp-Splat ranks the grasps proposed by GraspNet based on the grasp scores obtained from ASK-Splat. By leveraging the affordance score associated with each grasp pose, Grasp-Splat identifies grasp configurations that are more likely to succeed, depicted in Figure 9.

### C.2 Multi-Stage Robotic Manipulation

For multi-stage robotic manipulation, we begin by decomposing the manipulation task into stages. Our approach supports the specification of the stages comprising the task by a human or by a large

Without 3D Gaussian Infilling    With 3D Gaussian Infilling

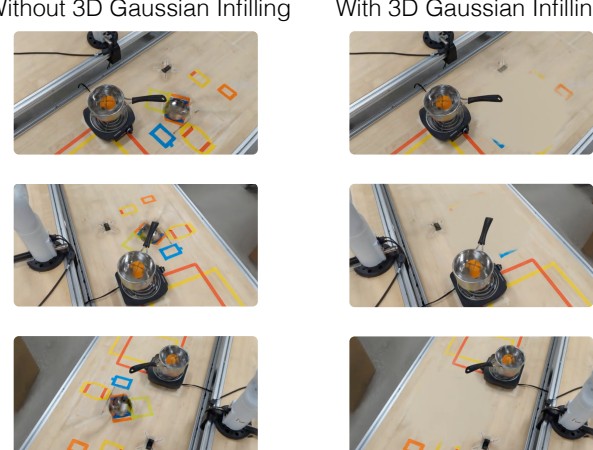

Figure 8: 3D Gaussian Infilling in SEE-Splat: (Left) In general, without 3D Gaussian infilling, transformation of the Gaussians (e.g., moving the saucepan from the table to the electric stove) introduces artifacts, such as the hole in the table after moving the saucepan. (Right) Via 3D Gaussian infilling, SEE-Splat generates photorealistic renderings of the edited scene.

GraspNet                Grasp-Splat

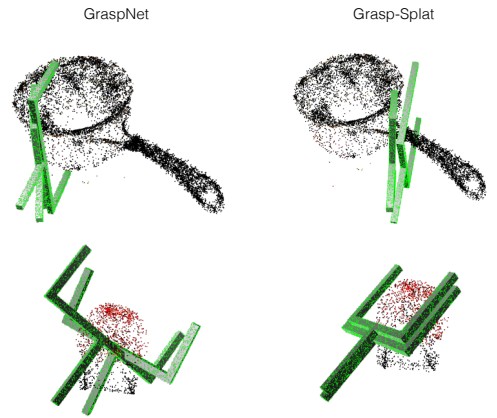

Figure 9: The top-two candidate grasps proposed by (left) GraspNet and (right) Grasp-Splat, leveraging the grasp affordance of each object. Qualitatively, the grasps generated by Grasp-Splat are more likely to succeed, compared to the grasps generated solely from Grasp-Net. Further, the grasps proposed by Grasp-Splat are better localized on the handle of the saucepan.

language model (LLM). In the case where the natural-language description of the task does not specify the stages involved in the task, we query an LLM for the stages required to complete the manipulation task. For each stage in the manipulation task, we utilize ASK-Splat, SEE-Splat, and Grasp-Splat to identify the relevant object and generate candidate grasp poses. Likewise, we query ASK-Splat for the target location for placing the object. We evaluate the feasibility of each candidate grasp using an off-the-shelf motion planner for the robotic manipulator, inputting the point cloud of the scene, extracted from ASK-Splat and SEE-Splat, into the motion planner, which the motion planner uses for collision detection during motion planning. We execute the top candidate grasps, moving on to the next if the robot motion planner fails to compute a solution to execute the selected grasp.

We execute the motion plan returned by the motion planner on the robotic manipulator. We note that the end-effector trajectory can be published to SEE-Splat for real-time visualization of the task in the virtual scene. In this case, we can apply the relative transformation between consecutive end-effector poses to the spatial attributes of the Gaussian associated with the object being manipulated. In

addition, we note that alternative approaches exist for computing the relative transformations of the object between consecutive frames. For example, if object-tracking information is available from sensors in the scene, SEE-Splat could leverage this information to update the spatial attributes of the Gaussians, rendering a video showing the real-time changes in the scene of the manipulator, including the motion of the object, as the manipulation task progresses. We proceed to the next stage in the manipulation task at the conclusion of the current stage, repeating the same procedures with the updated representation of the scene provided by SEE-Splat.

## Appendix D   Evaluations

We present additional experimental results of ASK-Splat, SEE-Splat, and Splat-MOVER in open-vocabulary, multi-stage robotic manipulation problems.

### D.1   Experimental Setup

#### D.1.1   ASK-Splat

We distill grasp affordances from the vision-affordance foundation model VRB [10], which is trained on the EPIC-KITCHENS dataset [41], consisting of videos of humans performing kitchen tasks, such as cutting fruits and vegetables. We note that the generalization of the affordance knowledge in ASK-Splat is limited by that of VRB, the underlying foundation model. VRB utilizes Language Segment-Anything (LangSAM) [42], which requires the specification of objects within each image for which it predicts the contact locations and motion direction after contact. This requirement is not limiting, in practice, as end-to-end object detectors that provide bounding boxes for all objects in the scene [43, 44, 45] could be used. We distill the grasp affordance scores from the heatmaps computed by VRB and the semantic embeddings from the vision-language foundation model `RN50x64`, CLIP-ResNet model [5]. We train the autoencoder and the Gaussian Splatting representation simultaneously, using the same set of images collected from a single scene, which takes about 25 minutes on an NVIDIA 3090 GPU with 24 GB of VRAM, not accounting for the one-time data-processing step to compute the affordance features prior to training the Gaussian Splatting model, which requires about 20 minutes. Computing the poses via structure-from-motion takes about 12 minutes. If the images are collected by a wrist-mounted camera on the robot, the structure-from-motion procedure can be omitted by directly leveraging the forward kinematics of the robot to determine the pose of the camera, reducing the total computation time required. We note that a similar training time for the model is required for the state-of-the-art baselines: LERF-TOGO [5] and F3RM [6]. To train ASK-Splat, we record a video of each scene using a smartphone and extract about 300 frames for training. The training data can consist of a fewer number of frames, depending on the spatial extent of the scene. We implement ASK-Splat in Nerfstudio [46], utilizing the training API available in Nerfstudio, using the sparse point cloud computed via structure-from-motion [47] for initialization.

#### D.1.2   Scenes

We consider only real-world scenes in our experiments, including a *Kitchen* scene (consisting of common kitchen cookware such as saucepans, chopping boards, and knives); *Cleaning* scene (consisting of common household cleaners such as disinfectant wipes, dish soaps, and cleaning sprays); *Meal* scene (consisting of cutlery such as plates, spoons, forks, and cups); *Random* scene (consisting of random items such as a pair of scissors, chess pieces, and keyholders); and a *Workshop* scene (consisting of tools such as a power drill, work mat, and scraper). Figure 10 shows these scenes. We note that the *Workshop* and *Random* scenes contain out-of-distribution objects with respect to the EPIC-KITCHENS dataset (i.e., objects not found in a typical kitchen), such as the power drill and the chess pieces.

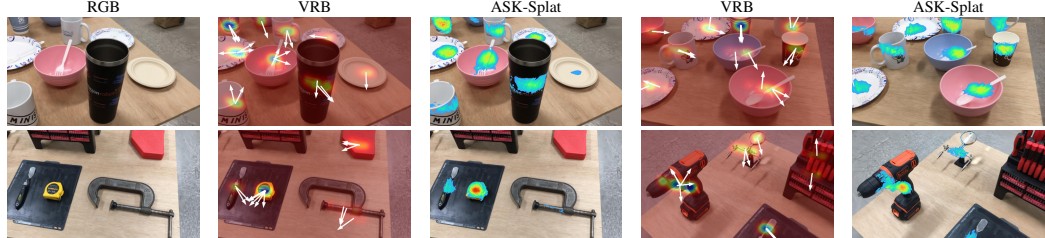

Figure 10: Grasp affordance for a *Kitchen* scene, *Cleaning* scene, *Meal* scene, *Random* scene, and *Workshop* scene (from top-to-bottom). We show the RGB image, grasp affordance as predicted by the vision-affordance foundation model (VRB), and the grasp affordance from ASK-Splat composited with rendered RGB images from ASK-Splat (from left-to-right).

### D.1.3  Splat-MOVER

We consider the multi-stage robotic manipulation task where the robot must sequentially pick and place two different objects and move them to a common goal location. The task is specified by a user that provides an open-vocabulary command, e.g., "Pick up the saucepan and move it to the burner, then pick up the lid and put it on the saucepan." For simplicity, we limit the task to two sequential pick-and-place maneuvers. However, we note that Splat-MOVER does not impose this limitation and is amenable to longer multi-stage manipulation tasks. Furthermore, we consider three adjacency goal location primitives ("on", "next to", and "inside") for the second object where each primitive is defined based on the geometry of the first object.

Specifically, we evaluate Splat-MOVER in four multi-stage manipulation tasks across three scenes: the *Kitchen*, *Cleaning*, and *Workshop* scenes. In the *Kitchen* scene, we consider a *Cooking* task where the robot is asked to place a saucepan on an electric burner (Stage 1) and subsequently place a fruit inside the saucepan (Stage 2). Further, in the *Kitchen* scene, we consider a *Chopping* task where the robot is asked to place a knife on a chopping board (Stage 1) and subsequently place a fruit next to the knife (Stage 2). We consider a *Cleaning* task (in the *Cleaning* scene), where the robot is asked to place a cleaning spray in a bin (Stage 1) and subsequently place a sponge next to the cleaning spray (Stage 2). Lastly, in the *Workshop* scene, the robot is asked to place a power drill on a work mat (Stage 1) and subsequently place a wooden block next to the drill (Stage 2), which we refer to as the *Workshop* task.

### D.1.4  Hardware Experiments

We implement Splat-MOVER in grasping and placing tasks on a Kinova Gen3 robot, equipped with a Robotiq parallel-jaw gripper. The Kinova robot is a 7-DoF robot with a maximum reach of 902 mm. We interface with the robot using the Robot Operating System (ROS), through which we send waypoints, which are tracked by the default low-level controllers provided by the robot. We utilize the MoveIt ROS package [48] for motion planning for the Kinova robot given a specified grasp pose. At each stage of the manipulation task, we extract a point cloud and a mesh from ASK-Splat and SEE-Splat, reflecting the progress in the task up to that stage, which we use as the environment representation within MoveIt for collision avoidance during planning.

### D.2  ASK-Splat Representation

We provide additional results showing the quality of affordance and semantic knowledge in ASK-Splat. In Figure 10, we show the RGB image, grasp affordance heatmap computed by VRB, and the grasp affordance heatmap rendered from ASK-Splat composited with the rendered RGB image.

We compute the Structural Similarity Index (SSIM) for each scene to assess the quality of the distilled affordance compared to the VRB-generated grasp affordance. The SSIM metric ranges between $-1$ (indicating greater dissimilarity) to 1 (indicating greater similarity). As expected, the Workshop scene

yields the smallest SSIM value of $0.592 \pm 7.20\mathrm{e}^{-2}$, recalling that the objects in this scene, such as the power drill and the scraper, are outside the training distribution of the VRB model. Nevertheless, the model shows relatively-good generalization performance, given that the grasp affordance region lies around the handle of the drill, shown in Figure 10 (bottom row). Likewise, the Meal scene achieves the highest SSIM score of $0.681 \pm 8.91\mathrm{e}^{-2}$, noting that the objects in the scene can be found in the dataset used in training the VRB model. Further, the Cleaning, Kitchen, and Random scenes achieve SSIM scores of $0.648 \pm 9.06\mathrm{e}^{-2}$, $0.647 \pm 1.30\mathrm{e}^{-1}$, and $0.614 \pm 8.37\mathrm{e}^{-2}$, respectively.

Figure 11 shows the semantic masks generated by ASK-Splat in the Cleaning scenes. Given a natural-language query, ASK-Splat localizes the relevant object in the scene based on the cosine-similarity of the Gaussians to the query. In Figure 11, ASK-Splat identifies the *flower*. However, the success of robotic manipulation tasks depend on the integration of semantic scene understanding with grasp affordance. As such, we show the semantic-affordance masks generated by ASK-Splat in Figure 11. With the semantic-affordance masks, a robot not only has the ability to identify a relevant object to grasp, the robot can also identify where on the relevant object to grasp.

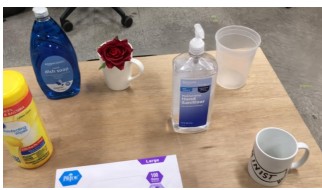 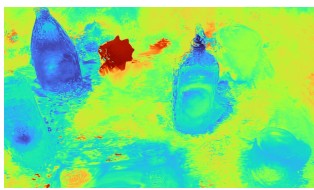 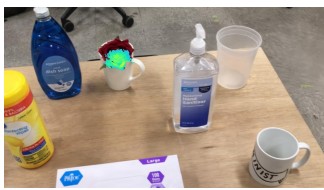

Figure 11: Affordance and Language Semantics in ASK-Splat: Given a natural-language query, ASK-Splat renders: (left) RGB images, (center) semantic masks of the scene, and (right) localized grasp affordance regions, for example, for a *flower* in the *cleaning* scene, evaluated at novel views in ASK-Splat.

### D.3  Splat-MOVER for Multi-Stage Robotic Manipulation

We examine the grasps proposed by Splat-MOVER compared to those proposed by prior work LERF-TOGO [5] and F3RM [6]. Figure 12 shows a few candidate grasps proposed by GraspNet, F3RM, LERF-TOGO, and Grasp-Splat for each of these objects. GraspNet does not consider the semantic features of the object in generating candidate grasps; as a result, the proposed grasps are not localized in regions where a human might grasp the object, unlike the candidate grasps proposed by F3RM, LERF-TOGO, and Grasp-Splat, which generate grasps closer to the handle of the respective objects. For example, the proposed grasps lie relatively close to the handle of the saucepan and the knife. Whereas LERF-TOGO and F3RM require guidance from a human operator or an LLM (in LERF-TOGO) or from a dataset of human demonstrations (in F3RM), Grasp-Splat does not require any external guidance to generate candidate grasps of similar quality, harnessing the grasp affordances provided by ASK-Splat.

### D.4  Comparison Between Splat-MOVER and Classic Geometric Representations

We neither perform a head-to-head comparison nor an ablation study against classic geometric representations, such as point clouds and voxels, noting the advantages of Gaussian Splatting (and more generally, radiance field) over these classic representations, among other reasons, which have been discussed in prior work, e.g., [7]. Here, we summarize a few of these advantages. Gaussian Splatting enables high-fidelity rendering in contrast to point-based representations, which suffer from missing geometry and aliasing, due to the limited resolution power of these methods. SEE-Splat leverages this high-fidelity rendering capability to provide a realistic digital twin for visualizing the evolution of the scene as the manipulation task unfolds. Moreover, Gaussian Splatting (and radiance fields in general) enables more effective language grounding in 3D, unlike point-based representations, as discussed in [26]. ASK-Splat exploits semantic distillation to enable semantic localization of the relevant objects given a language-guided task. Lastly, point-based representations

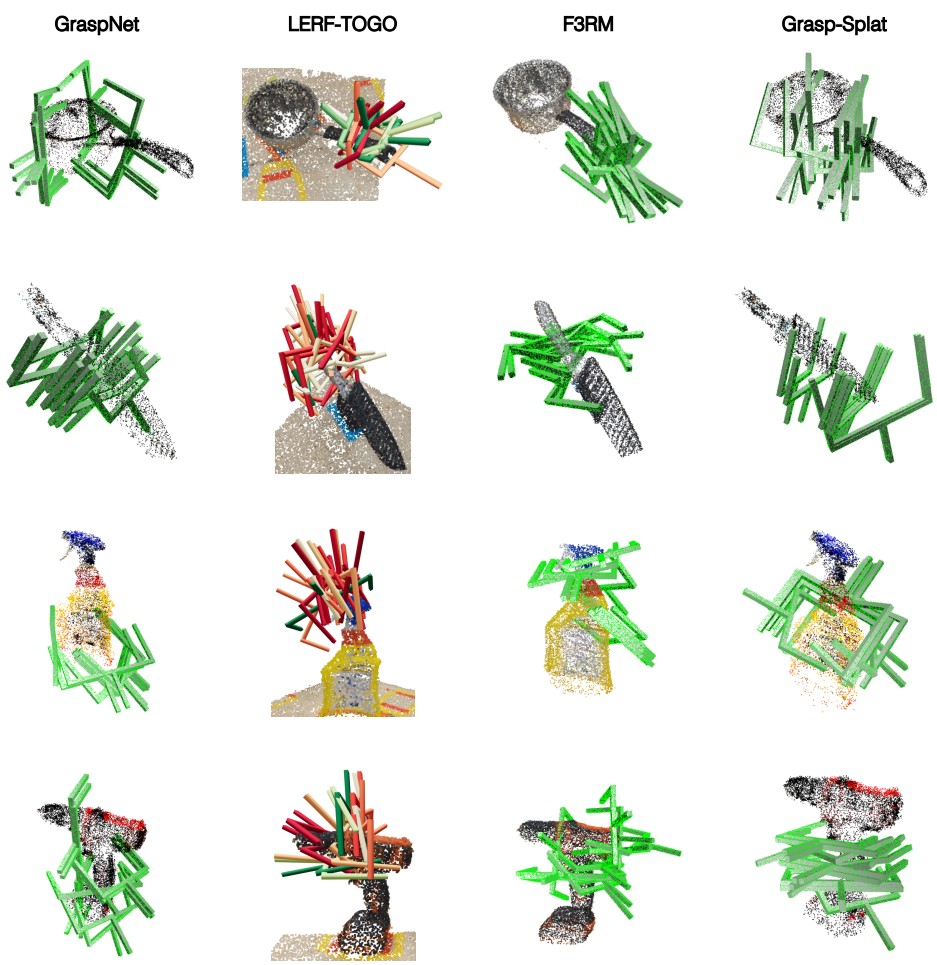

Figure 12: Candidate grasps for a *saucepan*, *knife in a guard*, *cleaning spray*, and *power drill* (from top-to-bottom), generated by GraspNet, LERF-TOGO, F3RM, and Grasp-Splat (from left-to-right). Although LERF-TOGO and F3RM require the specification of a grasp location from an operator, an LLM, or via human demonstrations to generate more-promising candidate grasps, Grasp-Splat generates candidate grasps of similar or better quality without requiring external guidance.

generally require a depth sensor to backproject a point cloud from each camera view, e.g., [49]. Our work does not require an RGBD sensor; the scene representation is constructed entirely from monocular images, which hinders a direct implementation of a point-based representation. Notably, under appropriate assumptions, Gaussian Splatting representations can be interpreted as a point-cloud, by reducing the ellipsoidal primitives to a point primitive given by the center of the ellipsoid. Hence, we can extract a point-cloud from a Gaussian Splatting representation, loosely demonstrating that the benefits of Gaussian Splatting encompass those of point-based representations.

### D.5 Comparison Between Splat-MOVER and Generalist Robot Policies

Generalist robot policies such as OpenVLA [15] and Octo [14] seek to enable versatile and adaptive behaviors across a wide range of tasks, environments, and natural language instructions. In contrast with specialized policies such as Diffusion Policy [50], these generalist policies are trained via imitation learning on mixtures of the Open-X Embodiment dataset [13]. We do not perform head-to-head comparisons with these methods for the following reasons:

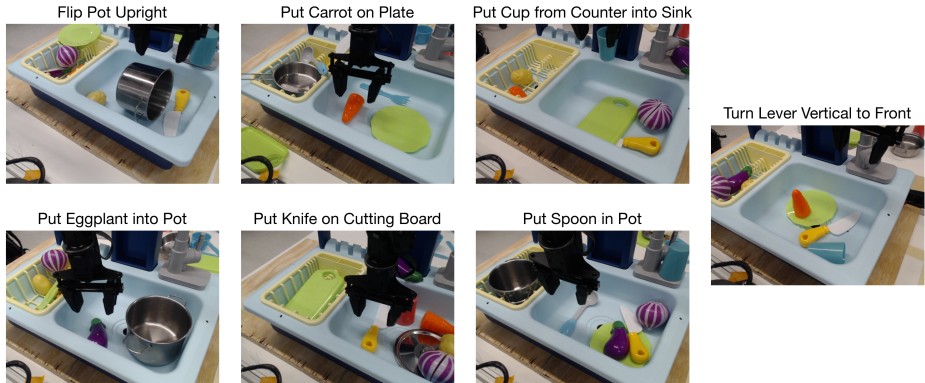

Figure 13: These images depict sample demonstrations from the original BridgeData V2 dataset. *This figure was taken from [15].*

Table 3: Success rates across different tasks and systems. *This table was taken from [15].*

| Task | RT-1-X # Successes | RT-2-X # Successes | Octo # Successes | OpenVLA (ours) # Successes |
|---|---|---|---|---|
| Put Eggplant into Pot (Easy Version) | 1 | 7 | 5 | 10 |
| Put Eggplant into Pot | 0 | 5 | 1 | 10 |
| Put Cup from Counter into Sink | 1 | 0 | 1 | 7 |
| Put Eggplant into Pot (w/ Clutter) | 1 | 6 | 3.5 | 7.5 |
| Put Yellow Corn on Pink Plate | 1 | 8 | 4 | 9 |
| Lift Eggplant | 3 | 6.5 | 0.5 | 7.5 |
| Put Carrot on Plate (w/ Height Change) | 2 | 4.5 | 1 | 4.5 |
| Put Carrot on Plate | 1 | 1 | 0 | 8 |
| Flip Pot Upright | 2 | 5 | 6 | 8 |
| Lift AAA Battery | 0 | 2 | 0 | 7 |
| Move Skull into Drying Rack | 1 | 5 | 1 | 5 |
| Lift White Tape | 3 | 0 | 0 | 1 |
| Take Purple Grapes out of Pot | 6 | 0 | 0 | 4 |
| Stack Blue Cup on Pink Cup | 0.5 | 5.5 | 0 | 4.5 |
| Put {Eggplant, Red Bottle} into Pot | 2.5 | 8.5 | 4 | 7.5 |
| Lift {Cheese, Red Chili Pepper} | 1.5 | 8.5 | 2.5 | 10 |
| Put {Blue Cup, Pink Cup} on Plate | 5 | 8.5 | 5.5 | 9.5 |
| Mean Success Rate | 18.5% ± 2.7% | 50.6% ± 3.5% | 20.0% ± 2.6% | 70.6% ± 3.2% |

1. These policies use very different data from ours, and are trained in quite a different manner from our method making a performance comparison problematic. They are based on VLM foundation models trained on vast data sets with vast resources. E.g., OpenVLA was fine-tuned for specific tasks using 8 A100 GPUs trained for 5 to 15 hours per task. This is just for fine tuning. Pre-training required full industrial scale foundation model training on vast internet-scale data in a large GPU cluster. Few academic labs have access to such resources. By comparison, our method was trained on a single 3090 desktop machine for 25 minutes.

2. These methods have access to real-time camera views (typically from multiple cameras) at execution time, while we do not require a camera at execution time. Conversely, we require a multi-view scanning phase prior to execution to build the map, while these policies require no such scanning phase, and do not use a map. Thus, it is difficult to make a fair comparison based on runtime data. While the real-time camera images enable robust control (e.g., to unmodeled disturbances) in generalist policies, the map enables global scene-understanding in Splat-MOVER, supporting a larger spatial context for task specification. Figure 13 depicts sample observations and tasks from the BridgeData V2 dataset used in fine-tuning generalist policies, showing that the goal and target object are always in frame, highlighting the relatively limited spatial-context required by these generalist policies.

3. Finally, these methods are, fundamentally, policies that are conditioned on images. Our method is not a policy. It is a scene representation and understanding framework. The actual motion policy we use is an off-the-shelf MoveIt implementation using a classical RRT planner. We do not claim this is the best option, or even a good option. Rather, the planning was not the focus of our contribution. Therefore, again, a performance comparison

is problematic. We instead compare with LERF-TOGO and F3RM, which are generally methods of the same kind of architecture as our method, using similar amounts of data in a similar fashion.

In lieu of a head-to-head comparison, we provide existing results on the performance of generalist policies, as stated in [15], where Octo, RT-1-X, RT-2-X, and OpenVLA are evaluated on a subset of the BridgeData V2 dataset. For each evaluation, OpenVLA and Octo are fully fine-tuned on a subset of 10-150 demonstrations for each task. The results for these evaluations, presented in Table 3, show that OpenVLA achieves a 70.6% average success rate across the 17 different tasks. This represents a 20 percent increase in average success rate over the next best generalist policy RT-2-X. While we recognize the strengths of these generalist policies, we do not consider them to be direct competitors to Splat-MOVER. Rather, we believe that integrating a generalist policy into the Splat-MOVER framework would result in a superior robotic manipulation system. Such a system would be compatible with a vast array of tasks specified over a large spatial range given natural language descriptions and robust to disturbances, e.g., in the object location, blending the global scene knowledge of Splat-MOVER with the closed-loop control of generalist policies.

## Appendix E   Related Work

**Open-World Robotic Manipulation.** Advances in foundation models [51] have enabled the development of open-world robotic manipulation methods, where robots manipulate objects given natural-language task instructions at runtime, without being trained on those specific objects or tasks. Existing methods fall into one of two categories: (i) ("smart policy") end-to-end visuo-motor policies based on a pre-trained large vision-language model [11, 12, 13], or (ii) ("smart map") methods that use a rich 3D scene representation that embeds features from a vision-language foundation model, interfacing with a traditional manipulation planning stack to drive robot motion. Our method is of the "smart map" variety. Two other recent methods also take this general approach: LERF-TOGO [5] and F3RM [6]. LERF-TOGO [5] builds upon the semantic NeRF representation in LERF [26], coupled with GraspNet [3] for grasp generation. Similarly, F3RM [6] uses the NeRF-based distilled feature field from [9] to embed task-relevant features learned from human demonstrations into the 3D scene model, which are used to generate and optimize candidate grasps. We compare against LERF-TOGO and F3RM as baselines.

Despite their effectiveness, these methods still face a number of challenges. LERF-TOGO [5] requires the specification of a grasp location on an object by a human operator or a large language model, which might not be readily available. Likewise, F3RM [6] requires human demonstrations, which might be difficult to collect. Both methods rely on NeRFs as the underlying scene representation, which can be time-consuming to train and is not easily edited to represent changes in the environment (e.g., due to manipulation actions by the robot). In this paper, we embed affordance features from a pre-trained 2D affordance model VRB [10] into the 3D scene, thereby avoiding the need for human demonstrations or human specification of grasp locations. We also build our scene on the Gaussian Splatting representation, which is faster to train and render than NeRF, and easier to edit to reflect object motions. We introduce a novel GSplat scene editing algorithm specifically designed for the tabletop, object-centric edits needed in robotic manipulation.

**Language-Embedded NeRFs and Gaussian Splats.** The core enabling technology behind the "smart map" variety of open-world manipulation methods is the neural feature field concept, which distills information from 2D data sources into a view-consistent 3D field by back-propagating through a differentiable image renderer. Neural Radiance Field (NeRF) [16] represents one of the earliest high-fidelity instance of this concept, distilling the 2D color information from RGB images into a 3D color and density field. NeRFs have been integrated into a number of robotics tasks, spanning navigation [17, 18], SLAM [19, 20, 21], and manipulation [22, 23]. Recent works have distilled semantic features from vision-language foundation models (such as CLIP [27], DINO [28], or LSeg [29]) into NeRFs [24, 25, 26], yielding a rich visual-semantic 3D scene representation. These methods were designed to highlight 3D semantic relevance to a text query (e.g., MaskCLIP [9] and LERF

[26]), or to produce 3D object masks for scene editing (e.g., CLIP-NeRF [25] and Distilled Feature Fields (DFFs) [24]).

Unfortunately, adoption of these rich representations in robotics is still hampered by two key challenges: (i) slow training and rendering times, and (ii) inability to reflect dynamic scenes. Gaussian Splatting [7], a recent photorealistic volumetric scene representation, addresses these fundamental limitations, by providing faster training and rendering speeds than NeRFs, while also offering the potential for modeling dynamic scenes through real-time scene editing, as we demonstrate in this paper. Gaussian Splatting represents the environment using 3D Gaussians, coupled with a fast rasterization-based image renderer. Some recent works [30, 31, 32, 33, 34] have explored embedding semantic features into 3D Gaussian Splatting, similarly to our work, but none of these have been used for robotic manipulation.

One key novelty in our method is that we distill grasp affordance features into the 3D field, together with CLIP semantic features. We use the pretrained VRB model [10], which infers a per-pixel grasp affordance metric from 2D RGB images, trained from videos of humans interacting with objects. Other works learn 2D grasp affordance models from still images ([52, 53, 54]) or from human-object interaction points in video data ([55, 56]), but none have distilled these 2D models into 3D fields to aid in robotic grasping. Both the semantic and affordance features in our model are crucial to grasp success, as shown in our experimental studies.

## Appendix F    Limitations and Future Work

Here, we discuss the limitations of Splat-MOVER in further detail. Splat-MOVER requires a one-time scanning procedure to collect RGB images of the scene, ranging from tens of images to a few hundred images. This procedure assumes that the images can be collected within the reachable workspace of the robot using a wrist-mounted camera. If the assumption fails to hold, a selfie-stick can be used to extend the reachable workspace of the robot to capture images over a wider area, as shown in prior work [6]. Moreover, future work will seek to integrate fast online Gaussian Splatting into ASK-Splat, eliminating the need for a brief scanning phase of the scene prior to training in addition to speeding up the training procedure. Additionally, we will seek to integrate sensor feedback into SEE-Splat to enable closed-loop, real-time scene editing, improving the accuracy of the scene representation. In this work, SEE-Splat does not model unexpected dynamic events such as slippage, which limits its robustness. Future work will seek to leverage online object pose tracking frameworks such as [57, 58] to estimate the pose of objects, enabling real-time updates to the underlying ASK-Splat scene to reflect the real-world. In addition, future work will seek to develop SEE-Splat into a fully-featured physics-based simulation engine, enabling more realistic reasoning about dynamic interactions between objects. Further, future work will seek to integrate robust robotic manipulation policies, such as [50, 15, 14], into Splat-MOVER to blend the robustness (and generality) of these policies with the scene-understanding and digital-twin capabilities of Splat-MOVER.

