# OpenReview forum: "Splat-MOVER: Multi-Stage, Open-Vocabulary Robotic Manipulation via Editable Gaussian Splatting"
_robot-learning.org/CoRL/2024/Conference — CoRL 2024_

### Official Review · Reviewer_JeoA · 2024-07-19
**Good paper but need refinement**

**Originality:** 4
**Technical Quality:** 3
**Clarity Of Presentation:** 3
**Potential Impact:** 3
**Recommendation:** 3
**Confidence:** 4

**Review:**

- strength:
  - The discussion of failure case and limitation is sufficient
  - This paper is clearly presented and easy to follow
  - Introducing the scene editing stage is novel and enables interesting downstream tasks.
- weakness:
  - Although it is understandable that authors have to cut down experiment section in the main paper due to page limitation, I suggest authors to make the method section more concise and expand more on the experiment section. Current experiment section is short and not thorough enough. I think authors could consider to break down the experiment section into several questions they want to answer.
  - Authors mention that F3RM requires collecting human demo in VR. I believe the demo can also be collected using robot's teacher's mode or other teleop interface. Authors might need to put more efforts into constructing baselines for fair comparisons.
  - For experiments, authors compare with LERF-TOGO and F3RM for grasping quality and success rate. Maybe authors want to include GaussianGrasper for comparison to be more convincing.
- comments:
  - This paper can be improved by introducing more tasks in the video, since the whole pipeline seems to use foundational models directly without more tuning.
  - It could be a minor issue, but the IK solver of the robot seems to be not optimal. For example, when grasping the pot, the robot end-effector finds a long path from its current position to the goal position.
  - I think more works can be included in "Open-World Robotic Manipulation". Here is a non-exhaustive list:
    - Wang, Yixuan, et al. "D $^ 3$ Fields: Dynamic 3D Descriptor Fields for Zero-Shot Generalizable Robotic Manipulation." arXiv preprint arXiv:2309.16118 (2023).
    - Ye, Jianglong, Naiyan Wang, and Xiaolong Wang. "Featurenerf: Learning generalizable nerfs by distilling foundation models." Proceedings of the IEEE/CVF International Conference on Computer Vision. 2023.
    - Qiu, Ri-Zhao, et al. "Learning generalizable feature fields for mobile manipulation." arXiv preprint arXiv:2403.07563 (2024).
    - Ke, Tsung-Wei, Nikolaos Gkanatsios, and Katerina Fragkiadaki. "3d diffuser actor: Policy diffusion with 3d scene representations." arXiv preprint arXiv:2402.10885 (2024).
  - In addition, StructDiffusion has similar scene editting process. Authors might consider compare them in the related works or experiments.
  - L237: Repeated image captions.

### Post-rebuttal comment
I keep my original rating. Overall, the authors answer my questions. However, the current experiment organization seems scattered and not well-organized.

**Quality Of The Limitations Section:**

3

**Questions For Rebuttal:**

- L38: As far as I understand, the scene should be represented as $\mu \in \mathbb{R}^{N\time 3}$, where $N$ is the number of Gaussian splats. Or you could say "each Gaussian is parameterized ..." Correct me if I am wrong.
- L208: Main paper and supplementary material mention that we could transform Gaussians. However, it is still not clear how this transformation is generated. Is it generated by LLM, human instructions, or some heuristics?

**Robotics Focus:**

4

**Summary Of Paper:**

This paper leverages Gaussian Splatting and foundational model to enable multi-stage manipulation tasks. Authors introduce three modules, including ASK-Splat, SEE-Splat, and Grasp-Splat to approach this problem.

**Summary Of Recommendation:**

This is overall a good paper. But it needs to modify its presentation and refine experiment section.

---

### Official Review · Reviewer_cpNM · 2024-07-22
**Review and comments on Splat-MOVER**

**Originality:** 2
**Technical Quality:** 3
**Clarity Of Presentation:** 3
**Potential Impact:** 3
**Recommendation:** 3
**Confidence:** 4

**Review:**

Overall, this paper show promising results of applying 3DGS-based representation for robotic manipulation tasks. The distillation of semantic feature and affordance features into 3DGS, utilizing scene editability of 3DGS in robotics are innovative. However, a few more points need to be improved in this paper:

The qualitative results of the feature distillation of Gaussian splatting used in this paper should be improved in the future.

The author should provide more justifications why 3DGS-based representation is chosen over classic geometric representation in the robotic manipulation task.

In real robot experiment, the author should also consider adding classic geometric baseline comparison, such as voxel-based representation.  In a relatively small scale environment, the requirement of 3DGS-based representation over pointcloud/voxel need to be justified in the experiments.

In the experiments, the author shows improved performance of NeRF-based baseline approaches, which is great. However, the authors are encouraged to add more explanations on why better results are achieved.

**Quality Of The Limitations Section:**

3

**Questions For Rebuttal:**

Provide more justification that 3DGS-based representation is chosen over classic geometric representation such as voxel/pointcloud. More experiments should also be conducted to provide justifications.

Provide explanations why 3DGS-based representation performs better than NeRF-based representations in the conducted experiments.

**Robotics Focus:**

4

**Summary Of Paper:**

This paper proposes Splat-MOVER, a 3DGS-based representation for open-vocabulary robotic manipulation.  It consists of ASK-Splat, SEE-Splat and Grasp-Splat.  ASK-Splat distills VLM features (CLIP) into 3GDS which enables open-set language query and scene/object understanding. SEE-Splat leverages the editability of 3DGS scene representations to enable multi-stage manipulation tasks. Grasp-Splat uses off-the-shelf grasping and motion planning modules to achieve manipulation. In the real-robot experiments, Splat-MOVER shows improvement comparing with the NeRF-based baselines.

**Summary Of Recommendation:**

This paper is among one of the earliest works using 3DGS with feature distillation in robotic manipulation tasks. Experiments show promising results comparing with NeRF-based baselines. However, more justifications need to be discussed to indicate why 3DGS representation is chosen and performs better than other representations in the particular robotic tasks.

---

### Official Review · Reviewer_wP4F · 2024-07-22
**Details in Appendix**

**Originality:** 4
**Technical Quality:** 3
**Clarity Of Presentation:** 2
**Potential Impact:** 3
**Recommendation:** 3
**Confidence:** 3

**Review:**

The overall quality of the paper is moderate to high. With 20 trials per task there is a clear improvement over baseline methods.

The idea of using the GS over NeRFs is an obvious win for robotics which this paper capitalizes on. However it is not necessarily true that GS is slower than NeRFs (i.e. mip-Nerf and zip-Nerf) especially when considering the presented work suggests training and auto-encoder in series (Appendix).

Strengths:
- An exciting use of GS for robotics with clear improvements over NeRF based models. This novel direction deserves more attention with regards to the context of robotics.
- Well written with clear notation.

Weaknesses:
- The a lot of the details were left to the appendix which was required to read to understand the method beyond the surface level. The appendix reads something like what the main paper should include.
- Would like to put the success rates into perspective compared to a smart-policy method that uses a foundational method as discusses in the early part of the related work.
- Not clear on how SEE-spalt models or reacts to un-expected dynamics such as slippage or errors in grasping.

The paper motivates the editability of GS but the evaluation appears limiting with regards to how this benefits long horizon tasks. This is partially due to the evaluation limiting to map/NeRF. Extending the evaluation to include smart-policies using an pre-trained open-vocab VLA model could help here (SayCan, RT-1 or more recently OpenVLA).

[Edit post-rebuttal]:
The the authors have reworked the structure and included more details in the main paper. The overall idea is interesting and novel. With this improved clarification the revised version I consider borderline accept to encourage more exploration of the GS approach to representations for robotics.

**Quality Of The Limitations Section:**

2

**Questions For Rebuttal:**

Please provide some additional details on data used for training the autoencoder to compress the semantic embeddings to 3 dimensions. Is this using only data from a single experimental scene? If not what training data is used? Is this slower than the GS optimization?

Practical challenges of collecting enough posed views are not well discussed. How many views are required and do the image poses extend beyond the reachability of a wrist mounted camera? If so please update the limitation section. Also how often would the scene need to be re-scanned in a multistage task. I.e.  under what conditions does the SEE-slpat not keep up with scene dynamics?

Do you have a data point for "smart-policy" methods on the evaluation tasks to better gauge the difficulty of the tasks beyond comparing to map based methods?

**Robotics Focus:**

4

**Summary Of Paper:**

The paper leverages the Gaussian Splatting representation with affordance and semantics to locate grasp points corresponding to a task expressed as natural language.

**Summary Of Recommendation:**

Scope of paper and/or division of content between main paper and appendix should be revised. Otherwise this is a great direction and could be highly impactful over the next two or more generations of papers in this direction.

---

### Author Rebuttal · Authors · 2024-08-14

In the following rebuttal, we provide the revised manuscript, comprising the main paper and the Supplementary Material. We have highlighted the major changes in blue, while typos and minor changes are described in the Official Comment to the reviewers, but are not highlighted in the manuscript.

---

### Decision · Program_Chairs · 2024-09-04

**Decision:**

Accept

**Comment:**

# Strengths
1. Utilizing the scene editability of 3DGS in robotics is novel.
1. The paper shows promising results for robotic manipulation tasks.
1. The real-world demonstrations showcase the effectiveness of the approach.
1. The paper is well-written.

# Weaknesses
1. The main body sometimes lacks necessary information about the method and experiments. Instead, the information is explained in Appendix. This is critical because the instructions for authors clearly state that "reviewers are not required to read the appendix."
1. Baseline comparison is insufficient.

### Post-rebuttal comment
Although the reviewers initially raised a few concerns, they agree that most of them have been addressed. As one of the reviewers pointed out, this paper could be one of the earliest works using 3DGS with feature distillation in robotic manipulation tasks. In the discussion between the Area Chair and the reviewers, they are in agreement on the quality and acceptance. I support their shared opinion.